# MODEL SPECIALIZATION FOR INFERENCE VIA END-TO-END DISTILLATION, PRUNING, AND CASCADES

## ABSTRACT

The availability of general-purpose reference and benchmark datasets such as ImageNet have spurred the development of general-purpose popular reference model architectures and pre-trained weights. However, in practice, neural networks are often employed to perform *specific*, more restrictive tasks, that are narrower in scope and complexity. Thus, simply fine-tuning or transfer learning from a general-purpose network inherits a large computational cost that may not be necessary for a given task. In this work, we investigate the potential for *model specialization*, or reducing a model's computational footprint by leveraging task-specific knowledge, such as a restricted inference distribution. We study three methods for model specialization—1) task-aware distillation, 2) task-aware pruning, and 3) specialized model cascades—and evaluate their performance on a range of classification tasks. Moreover, for the first time, we investigate how these techniques complement one another, enabling up to $5\times$ speedups with no loss in accuracy and $9.8\times$ speedups while remaining within 2.5% of a highly accurate ResNet on specialized image classification tasks. These results suggest that simple and easy-to-implement specialization procedures may benefit a large number practical applications in which the representational power of general-purpose networks need not be inherited.

## 1 INTRODUCTION

The success of deep learning has been enabled by the availability of large reference datasets capturing general-purpose predictive tasks. In turn, these reference datasets are routinely accompanied by reference architectures and pre-trained model weights that achieve state-of-the-art accuracy on these general-purpose tasks. As a result, practitioners of deep learning often build their machine learning applications by obtaining a pre-trained, general-purpose model and tailoring it to their specific task and data distribution (e.g., via transfer learning, or fine-tuning (Yosinski et al. (2014); Razavian et al.)). Almost all deep learning software frameworks have a tutorial for performing this procedure.

While this task-specific fine-tuning of general-purpose models frequently results in highly accurate task-specific models, the resulting task-specific models inherit the performance overheads required by the original, more general task. For example, if we obtain the weights for ResNet-152 (He et al., 2016) pre-trained on ImageNet 1000-class classification dataset, then fine-tune the last layer for a binary classification task, our fine-tuned model will inherit the high-quality featurization provided by ResNet-152. However, our fine-tuned model will also inherit ResNet-152's runtime overhead, permitting approximately 286 image classifications per second on an NVIDIA P100 GPU (and 2343 images/second for the more modest ResNet-18). In contrast, as we illustrate in this paper, a model architecture specifically tailored to this task could achieve comparable accuracy but with inference performance of over 11,763 images/second on the same hardware.

Thus, in this paper, we investigate the potential of *model specialization*, or improving a model's inference performance by leveraging task-specific knowledge such as a limited test distribution. In contrast with techniques such as model distillation (Hinton et al. (2015)) or compression (Han et al. (2015)) that produce inference-efficient models for the *same task* as a reference model, model specialization produces an inference-efficient model for a *particular, restricted* task. Because the specialized task is often much simpler than many benchmark tasks upon which models are trained (e.g., cat/dog vs. ImageNet top-5), this specialized model can be much smaller while still retaining high accuracy for the specialized task. A handful of recent studies have independently explored

several mechanisms for specialization, including cascades (Kang et al., 2017; Bolukbasi et al., 2017; Shen et al., 2017) and pruning (Molchanov et al., 2017). However, it is unclear how these methods interact, whether they can be combined, and to what extent they complement one another. In response, here, we combine these methods for the first time, leveraging distillation, pruning, and cascade techniques in a single, end-to-end training and inference pipeline.

Our main contribution is to systematically evaluate and combine three techniques for model specialization: task-aware distillation, pruning, and cascades. Task-aware distillation uses a large, general-purpose reference network to produce task-specific training data to train a smaller, task-specific network than would be possible for the general task. Task-aware pruning removes network filters and weights from a network based on task-specific data distribution, allowing more pruning than general-purpose weight or filter pruning. Specialized model cascades combine efficient specialized networks with less efficient reference networks to recover accuracy lost in network shrinkage and to enable fine-grained trade-offs between speed and accuracy. As it is unclear how these three techniques interact across specialization tasks, we affirmatively show that these three techniques can be combined into a task-specific model training pipeline. In our proposed pipeline, a model is automatically distilled, then pruned, and cascaded, thus producing a specialized model that can be as accurate as a general-purpose model for the given task but more efficient for inference.

We present results in two settings, in which we have access to a large pool of unlabeled data, and in which we have access to a small amount of labeled data. Compared to conventional transfer learning, the above specialization pipeline demonstrates empirical speedups of up to $5\times$ over ResNet-18 with no loss in accuracy, and up to $9.8\times$ while maintaining binary classification accuracy within $2.5\%$. Our results demonstrate that each mechanism for specialization is independently valuable, and that combining these methods yields improvements unavailable to each mechanism in isolation. Thus, these results suggest that, for many applications, the neural network inference overheads can be substantially improved over conventional transfer learning using a combination of conceptually simple and easy-to-implement distillation, pruning, and cascade-based specialization methods.

## 2 RELATED WORK

As neural networks are increasingly deployed, improving the runtime performance of model inference is an increasingly active area of research, with several related efforts spanning distillation, cascades, and pruning.

Several network architectures for efficient inference have been developed, ranging from MobileNets (Howard et al. (2017)) to ShuffleNets (Zhang et al. (2017)). These network architectures exploit information about the inference setting (e.g. power constraints or hardware) to perform efficient inference. However, they do not take advantage of *task-specific* knowledge. Thus, we view these architectures as complementary to our goal of model specialization.

In model compression and pruning (Han et al., 2015), parts of the network are removed for improved efficiency, including use of fewer parameters to improved inference efficiency. Model pruning at the weight level typically requires specific hardware to achieve inference speedups (Han et al. (2016)). However, pruning at the filter level can give speedups on existing hardware (Li et al. (2016); Molchanov et al. (2017)).

Similar to Li et al. (2016) and Molchanov et al. (2017)), we investigate pruning at the filter level to achieve speedups on existing hardware. Unlike Li et al. (2016), we exploit *task-specific* knowledge to prune networks further than task-agnostic methods. Perhaps closer in spirit, Molchanov et al. (2017)) leverage dataset-specific pruning methods using AlexNet and VGG-16. In our study of model specialization, we extend these methods to prune residual networks, which empirically result in higher accuracy.

In model distillation (Hinton et al. (2015)), a teacher model is used to train a student model to mimic the teacher model on the *same task*). In our study of specialization, we adapt this idea, but perform distillation in a task-aware manner by restricting the data distribution that participates in distillation.

Model cascades (Viola & Jones, 2001)—in which a sequence of increasingly computationally expensive models are used for inference—are increasingly popular in deep network inference. Specifically, (Bolukbasi et al., 2017; Wang et al., 2017) provide mechanisms to train deep networks

(or cascade structures) whose depths are adaptive and vary according to the difficulty of input. NoScope (Kang et al. (2017)) performs model specialization for binary object detection in video, combining cost-based model search with a three-stage cascade of models that detect differences in video and execute models specialized for a given video and query. Similarly, Shen et al. (Shen et al., 2017) develop methods for configuring a two-stage cascade of deep models for video trained online, with a fixed model architecture for the specialized model.

Our work builds upon these studies by providing an empirical analysis combining cascades with two complementary means of specialization—task-specific distillation and pruning. We are unaware of existing methods for combining specialization via distillation, pruning, and cascades, or analyzing the extent to which these techniques are complementary across a range of classification tasks.

## 3 MODEL SPECIALIZATION TECHNIQUES

In this section, we present an overview of three mechanisms for model specialization: task-aware distillation, task-specific pruning, and specialized model cascades. While these techniques have been previously explored independently in several settings, our goal is to provide a means of combining them, which we present at the end of this section.

### 3.1 TASK-AWARE DISTILLATION

In task-aware distillation, a smaller and more efficient model is trained to imitate a reference general-purpose model's behavior on a specific task. In contrast, in conventional model distillation, the student model attempts to imitate the parent model on the *same task* (e.g. 1000-way ImageNet).

To perform task-aware distillation, we assume the presence of a large pool of unlabeled data (e.g. gathered from the web) drawn from a restricted target distribution corresponding to a specific task (e.g., "cat" vs "dog"). Given this unlabeled data, we can distill a general-purpose parent model into a much smaller, inference-efficient child model that approximates the distribution of interest. Specifically, in distillation, we select a smaller model architecture, and train from scratch on the data that is labeled by the parent model. We find that, in some instances, the child model can outperform the parent model in both inference efficiency and in accuracy. In other instances, the child model is less accurate compared to the parent model, but this accuracy gap can be rectified using model cascades (described below).

In our experiments, we use the YFCC100M dataset (Thomee et al. (2016)), a pool of 100M images from Flickr as our unlabeled pool of data. We use a pretrained ResNet-152 for labeling the images and use the top prediction as ground truth for the child model.

### 3.2 TASK-SPECIFIC PRUNING

In task-specific pruning, we use the activations from the neural network to prune unnecessary filters in convolutional layers. Much of the prior literature focus on task-agnostic methods for pruning that use the weights of the neural networks to decide when to prune (Li et al. (2016)). Compared to task-oblivious methods, we show task-specific pruning can eliminate up to 63% of the filters compared to 10.8% in the general case (Li et al. (2016)). In contrast to prior works that also prune in a task-specific manner, we focus on ResNets, which are more accurate than networks without skip connections.

As previously mentioned, to achieve our goal of fast inference, we prune networks at the *filter* level. This is in contrast to prior work that prunes at the individual weight level. While specialized hardware can take advantage of weight-level pruning (e.g., (Han et al., 2016)), current deep learning frameworks and GPUs do not currently accelerate sparse networks.

Unlike task-aware specialization, we find that task-specific pruning does not require access to a large unlabeled pool of data and can be accomplished with relatively small amounts of labeled data.

Using the notation of He et al. (2016), residual networks can be seen as blocks of the form

$$y_i = \mathcal{F}(x_i, \{W_i\}) + x_i$$

where $W_i$ are weights of a convolution $c_i$, and $\mathcal{F} = \sigma(c_i * x_i)$ for some non-linearity $\sigma$.

In the 2D case, $W_i$ can be seen as a vector of *filters*, namely $W_i = w_i^{(k)}$. In this work, we prune at the filter level for efficiency purposes. To select which filters to prune, we evaluate the network of a subsample of the data (5-10 batches), $x$ and compute the activations output by $c_i * x_i$. If we denote the activations $a_i = c_i * x_i$, we rank the filters by the $l_1$ norm:

$$\Theta(a_i) = \frac{1}{n} \sum_k |a_i^{(k)}|$$

The filters are ranked according to their activations, and we eliminate the filters with the lowest rank, taking care not to fully eliminate any layer. We zero-pad as necessary to maintain the dimensions.

### 3.3 SPECIALIZED MODEL CASCADES

To provide fine-grained trade-offs between speed and accuracy, we compose the specialized model and a general-purpose reference model in a *specialized model cascade*. In a specialized model cascade, a specialized model defers to the reference model when it is unsure about a given prediction.

As we have discussed, there is a large existing body of work on model cascades to (Viola & Jones, 2001) and more recently in (Cai et al., 2015; Shen et al., 2017; Kang et al., 2017; Wang et al., 2017). We borrow ideas from these techniques and others but we note a few key differences. First, much of the prior work on cascades focuses on cascades to retain a specific accuracy by exploiting class imbalance. In this work, we explicitly explore the trade-off between accuracy and throughput, and allow specialized models to output predictions directly.

In its full generality, the problem of deciding when to defer to a larger model is as difficult as the prediction task. However, we empirically observe that simple heuristics such as placing a hard cutoff on specialized model confidence do well in practice, achieving up to $5\times$ speedups within 0.1% accuracy of a reference model.

If we denote the cost of querying the specialized model $S$, the cost of of querying the reference model $R$, and the fraction of data points that the reference model is queried on as $f$, the total cost of querying the cascade is $S + fR$. Therefore, if $f < 1 - \frac{S}{R}$, the cost of querying the cascade is less than the cost of the reference model. We show empirically that $f$ is often small enough to warrant the use of cascades for specialized tasks.

To choose when to query the reference model, we use a simple classifier, placing a linear threshold on the distance between highest and second-highest class predictions. We show that this simple heuristic performs well in practice.

### 3.4 COMBINED, END-TO-END SPECIALIZATION PIPELINE

While each of the above specialization techniques have been investigated in isolation, we propose to evaluate how they can work in concert. To combine these techniques, we propose a three-stage pipeline. First, provided sufficient unlabeled task-specific training data is available, distill a specialized model that is smaller than a reference. Second, prune the distilled model by observing its activations on the task-specific training data. Finally, construct a cascade betwee the resulting distilled and pruned model, parametrizing the cascade threshold to attain the desired accuracy and inference speed.

As observed in the literature, neural networks are over-parameterized (Zhang et al., 2016) and can be pruned with limited loss in accuracy. However, it is not immediately clear that models that have been distilled for a particular task also exhibit this property. In the next section, we illustrate that these specialized networks can be pruned even further, thus providing modest speedups. In addition, the above procedure raises several questions regarding model architecture search, pruning procedure, and cascade configuration. We report results across several classification tasks in the next section.

## 4 EXPERIMENTAL EVALUATION

We evaluate our proposed model specialization pipeline in the context of image classification and report results from distillation, pruning, and cascades both independently and in concert.

## 4.1 EXPERIMENTAL SETUP AND PROCEDURES

We report experiments using PyTorch v0.2 (the latest at the time of writing), with the exception that we generate labels for the YFCC100M dataset were generated with TensorFlow. We run all experiments on an NVIDIA P100 GPU with 16GB of memory and, to standardize the network architecture, use variants of ResNets (He et al., 2016) for all experiments.

To measure inference performance of a given model, we loaded the model into the GPU and "warm-started" the model by performing inference over 100 batches of data, which allowed the GPU kernels to be registered. We then ran 20 batches over the data and reported the mean inference time. To find the optimal batch size, we ran the preceding procedure over batch sizes in powers of two until the GPU could no longer fit the data in its memory.

For training, we used SGD with momentum of 0.9, weight decay of 1e-4, and a batch size of 32. We varied the initial learning rate across experiments, but generally started at 0.1. For data augmentation, we randomly horizontally flipped each image and cropped the image randomly from 8% to the full size, and then resized it to the appropriate input size (either 224 for pre-trained models or 65 for our specialized models).

We have anonymized our code for double-blind review and will release as open source following.

## 4.2 TASK-AWARE DISTILLATION AND SPECIALIZED MODEL CASCADES

To begin, we evaluate task-aware distillation and specialized model cascades, as cascades provide a means of smoothly trading off between speed and accuracy.

We evaluate task-aware distillation in the context of both binary image classification and multi-way classification. We perform validation using the ImageNet validation set, containing at least 500 images per target class. We picked five animal classes (bird, dog, feline, insect, snake) and one extra class (car) from the WordNet hierarchy and specialized models to perform binary classification for each pair. In addition, we trained a separate model to perform 6-way classification. As an unlabeled set of training data for distillation, we use the YFCC100M dataset (Thomee et al. (2016)), containing 100M images from Flickr as an unlabeled pool of data. To select images from specific target classes, we used ResNet-152 to perform classification on YFCC100M.

We trained distilled models with the following architectures: input resolution of 65, 16 filters in the first layer with filter doubling, and padding to align the layers, and trained networks with 10, 18, and 34 layers. We denote these networks "TRNX" where $X = 10, 18, 34$. Thus, compared to conventional ResNets, these "Tiny" ResNets have smaller resolution (i.e., 65 x 224 pixels) and smaller number of filters (e.g., 16 vs. 64 filters), selected via manual architecture search. We used SGD with an initial learning rate of 0.1 and momentum of 0.9. We decreased learning rate by a factor of 10 for every epoch that the validation loss did not decrease, and ran at most 50 epochs, with early stopping if the validation loss or accuracy did not increase for 5 epochs. After this training, we fine-tuned the models for 10 epochs on the corresponding subset of ImageNet training data.

ResNet-152 substantially outperforms smaller networks such as ResNet-18 on ImageNet (e.g., 97.0% vs 95.6% F1 on insect vs snake, 98.4% F1 vs 97.2% F1 on multi-way classification), so we cascade our specialized models with ResNet-152. We use a simple heuristic to select when to query the reference model: we use a hard threshold based on the difference between the top two confidence values. This allows a smooth trade-off between accuracy and inference speed.

We show three results in Figure 1, illustrating the smooth trade-off between accuracy and inference-speed. In the best binary classification result (bird vs. dog), the simple combination of distillation and cascading can achieve a $5.2\times$ speedup over ResNet-152 while maintaining accuracy within 0.5 F1 points. In our worst binary classification observed result (insect vs. snake), we can achieve a $1.9\times$ speedup while maintaining accuracy within 0.5 F1 points of ResNet-152.

Additionally, in the multi-class classification task, we observe a $4.1\times$ speedup while staying within 0.5 F1 points of ResNet-152. While staying within 1 F1 point of ResNet-152, we observe a $5.6\times$ speedup, demonstrating a similar trade-off between speed and accuracy. Thus, by modifying the cascade cut-off for the specialized model, we allow users to navigate a smooth and configurable trade-off between speed and accuracy that can span orders of magnitude in computational efficiency.

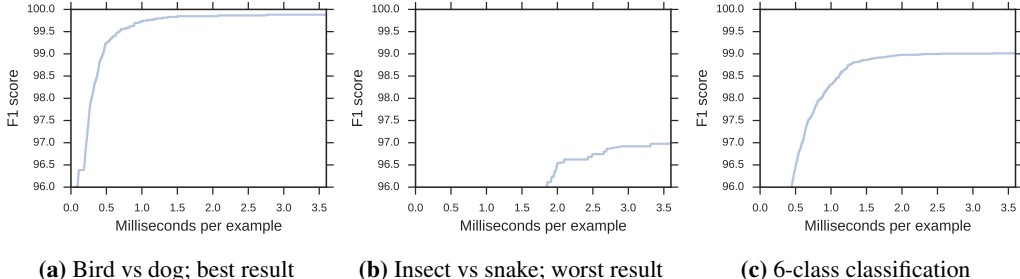

**(a)** Bird vs dog; best result     **(b)** Insect vs snake; worst result     **(c)** 6-class classification

**Figure 1:** Speed versus accuracy for specialized model distillation combined with ResNet-152 in a cascade.

| Model | Pass through rate (99.5%) | Speedup (99.5%) | Speedup (99.7%) |
|-------|---------------------------|-----------------|-----------------|
| TRN8  | 33%   | 3.89× | 3.51× |
| TRN10 | 14.8% | 5.39× | 5.01× |
| TRN18 | 11%   | 3.63× | 2.38× |
| TRN34 | 7.8%  | 2.15× | 1.93× |

**Table 1:** Speedup of different specialized model cascades over ResNet-18 at the 99.5% and 99.7% accuracy mark (baseline of ResNet-18 achieves accuracy of 99.8%).

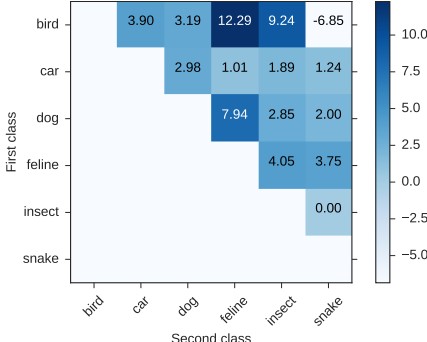

**Figure 2:** F1 score differential between training from scratch and distillation on ImageNet using TRN34.

| Model | Train from scratch F1 | Specialized F1 |
|-------|-----------------------|----------------|
| TRN10 | 76.6% | 89.5% |
| TRN18 | 81.0% | 89.9% |
| TRN34 | 80.1% | 89.9% |

| Model | Train from scratch accuracy | Specialized accuracy |
|-------|-----------------------------|----------------------|
| TRN10 | 93.8% | 97.3% |
| TRN18 | 95.9% | 97.7% |
| TRN34 | 95.8% | 97.9% |

**(a)** Specialized model accuracy compared to training from scratch on the 6-class classification task

**(b)** Specialized model accuracy compared to training from scratch on the Kaggle cat vs dog dataset.

**Table 2:** Specialized model performance on classification tasks

We also evaluated task-aware distillation using the Kaggle cat vs dog dataset (Elson et al. (2007)), containing 25,000 images (evenly split between cats and dogs), of which 1,000 were used as a validation set. We repeated the same setup as above, except we used the Kaggle dog vs cat dataset for fine-tuning and testing, and we used a ResNet18 as the baseline because this model already achieved an accuracy of 99.8%. We show the speedups achieved in Table 1 at two accuracy levels: 99.5% and 99.7%. We can see that even when compared to a smaller model (ResNet18 vs ResNet152), specialized models can achieve up to a 5× speedup with only a 0.1% loss in accuracy.

To determine whether task-aware distillation assists with training, we compared against networks trained from scratch on the ImageNet and Kaggle cat vs dog datasets, respectively, without using the larger pool of unlabled data from YFCC100M. We used the same architecture and training procedure as when training on the YFCC100M dataset. As Figure Figure 2 shows, task-aware distillation outperforms training from scratch in nearly every instance, and similar trends hold for other architectures. Similarly, task-aware distillation outperforms training from scratch on the 6-class

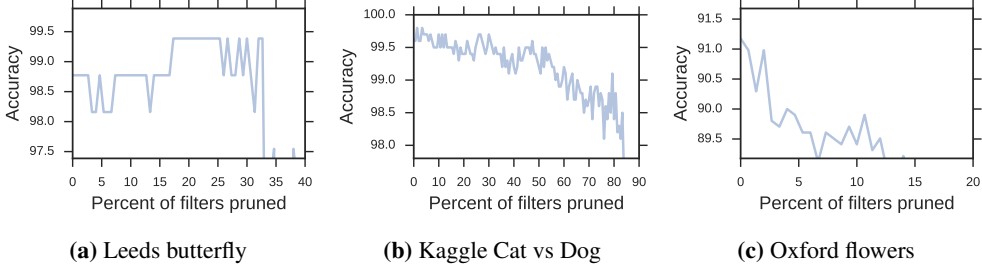

**Figure 3:** Pre-trained ResNet-18 filters pruned via task-specific pruning versus accuracy.

| Task | Percent pruned | Accuracy (delta) |
|------|---------|---------|
| Leeds Butterfly | 61.3% | -0.6% |
| Cat/dog | 53.3% | -0.3% |
| Oxford Flowers | 4.6% | -1.5% |

**(a)** Pre-trained ResNet-18 filters pruned via task-specific pruning vs accuracy.

| Task | Speedup | Accuracy (delta) |
|------|---------|---------|
| Leeds Butterfly | 33% | -0.6% |
| Cat/dog | 9% | -1.0% |
| Oxford Flowers | 2% | -0.1% |

**(b)** Observed GPU speedup vs accuracy via task-specific filter pruning of pre-trained ResNet-18.

**Table 3:** Accuracy and runtime versus proportion of filters pruned for ResNet-18.

| Percent specialized network pruned | Cascade pass-through rate | Accuracy (delta) |
|------|------|------|
| 17.8% | 11.8% | -0.1% |
| 27.8% | 7.2% | -0.3% |
| 33.3% | 0% | -2% |

**Table 4:** Specialized model pruning with cascade pass-through rate at varying accuracies.

classification task. We observe similar trends for the Kaggle cat vs dog dataset in Table 2b, where the distilled models consistently outperformed the models that were trained from scratch. This indicates that using a large pool of unlabeled data from the task distribution can improve accuracy.

## 4.3 TASK-AWARE PRUNING

We also evaluated task-aware pruning in the context of classification. Since task-aware pruning starts with a pretrained model, it does not require a large pool of unlabeled data. Thus, we evaluated task-aware pruning with smaller datasets: Kaggle cat/dog (binary classification with 24K training, 1K validation images), Leeds butterfly (10-class butterfly classification, 669 training, 163 validation images) (Wang et al. (2009)), and Oxford Flowers 102 (102-class flower classification, 1020 training, 1020 validation images) (Nilsback & Zisserman (2008)).

As a baseline for these experiments, we consider a pre-trained ResNet-18 fine-tuned to each dataset. We pruned 32 filters per iteration and trained the network for 3 epochs after every pruning iteration. Figure 3 illustrates the results. The difficulty of the task affects how much of the network can be pruned. As shown in Table 3a, in the best case, 2944 out of the 4800 filters (61% of the network) can be pruned with a 0.6% decrease in accuracy. However, with a more difficult dataset and fewer labeled examples (Oxford-102), only 5% of the network can be pruned with minimal loss in accuracy.

We observe modest gains in efficiency as shown in 3b. We defer the full discussion of this effect to Appendix A, but, briefly note that preserving the identity connections in ResNet architectures requires memory operations that are not well supported in existing deep learning frameworks. We suspect hand-tuned GPU kernels would ameliorate this problem.

## 4.4 END-TO-END FACTOR ANALYSIS

Finally, we evaluate the extent to which task-aware distillation, task-aware pruning, and specialized model cascades are complementary and can be combined in a single training pipeline as proposed in Section 3.4. First, we performed task-aware distillation using the YFCC100M dataset as above. We

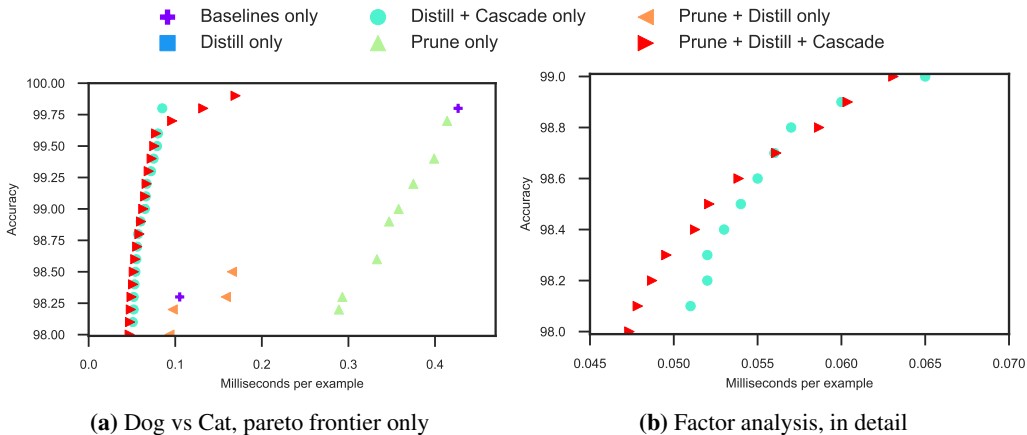

**(a)** Dog vs Cat, pareto frontier only      **(b)** Factor analysis, in detail

**Figure 4:** Factor analysis for model specialization techniques comparing speed and accuracy.

pruned the distilled networks using the procedure above and varied the number of filters pruned per step so each step pruned approximately 1% of the original network.

We compared to baselines of fine-tuned ResNet-18 and fine-tuned AlexNet (Krizhevsky et al. (2012)). We also compared to ShuffleNet (Zhang et al. (2017)) and MobileNet (Howard et al. (2017)) but found they were slower than ResNet-18 on our target GPU environment (and, compared to ResNet-18 on GPU, slower on our server-class CPUs). This is unsurprising because these networks were designed for inference on resource-constrained CPUs, so we omit results for ShuffleNet and MobileNet.

Figure 4 demonstrates the result. We observe that the specialized model cascades coupled with task-aware distillation can give a $5\times$ speedup with a 0.1% loss in accuracy compared to ResNet-18, and an $8.9\times$ speedup with a 2.5% loss in accuracy compared to ResNet-18. Compared to fine-tuning AlexNet or ResNet-18, specialization allows substantially higher throughput for a given accuracy level, and the ability to cascade models allows fine-grained trade-offs between accuracy and speed.

As we show in Table 4, we can prune 33% of the distilled network and achieve comparable accuracy. However, as in our pruning experiments, we observed only modest speedups by enabling pruning in addition to distillation and cascades. There were two regimes in particular where enabling pruning assisted. First, for high accuracy levels, pruning slightly boosted accuracy compared to distillation, possibly acting as a regularizer. Second, for lower accuracy levels (inset,Figure 4b), pruning delivered speedups of up to 8%.

These results suggest that three-stage model specialization can deliver tangible benefits with configurable impact on accuracy. For specialization can deliver up to $5\times$ speedups with comparable accuracy (within 0.1%), and, should applications permit accuracy degradation, even larger performance improvements (above, with 5% loss in accuracy, $10.1\times$ faster).

## 5 CONCLUSIONS

While the availability of large, general-purpose pre-trained networks provides an attractive basis for highly accurate inference via fine-tuning, wholesale adoption of these network architectures imposes a large inference performance penalty for specific tasks that may not be warranted. In this work, we investigated the interaction of three simple but powerful techniques for *model specialization*, or improving the inference performance of a neural network for a given task. By combining task-specific distillation, pruning, and cascades, we demonstrated empirical speedups on the GPU of up to $5\times$ over ResNet-18 with no loss in accuracy, and up to $9.8\times$ while retaining accuracy within 2.5%.

Given that each method is individually simple and easy to implement and are complementary when performed together, we believe that our proposed end-to-end specialization pipeline may be useful for circumventing a large fraction of computational overhead in many practical uses of deep networks.

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

## A    ADDITIONAL DISCUSSION

**Intuition for model specialization**    The over-parameterization of deep neural networks (Zhang et al. (2016)) has been observed in prior studies ranging from distillation (Hinton et al. (2015)) to network pruning (Han et al. (2015)), largely for general-purpose predictive tasks. In this work, we study means of reducing this over-parameterization in a task-aware manner.

As informal intuition, suppose we have a reference classifier $R : X \rightarrow Y$. In model specialization, we train a model $S : X' \rightarrow Y'$, where $(X', Y')$ is a marginal of $(X, Y)$ (e.g. cat vs dog compared to the full ImageNet-1000 task). In this case, $R$ can be used to generate labels given samples from $X'$. Moreover, depending on the task complexity of $R$ versus $Y$, there may be a simpler classifier (or larger set of classifiers; e.g., set of hyperplanes separating the classes) that translates to a less computationally intensive model. A simple example of this task is building a classifier to identify points in four quadrants of a 2D plane. A linear classifier cannot reliably separate these four quadrants, but many linear classifiers can reliably separate any two quadrants. Thus, the model specialization procedures we study here can be considered to implicitly leverage the structure of task-specific marginals to obtain computationally efficient representations.

**Efficiency of pruning**    In CNNs without identity connections such as VGG, removing filters from convolutional units is straight-forward and can immediately give performance gains (Molchanov et al. (2017)). However, in ResNets and similar network architectures (e.g. dual-path networks (Chen et al. (2017))) with multiple paths, removing filters is not as straight-forward. At every junction, the size of all inputs must match. As a result, in our pruned networks, we "fill in" any gaps with zeros. This results in complicated memory access patterns on GPUs, which is why we observe only modest speedups of our pruned networks. We believe that hand-tuned GPU kernels or more careful memory access could provide additional speedups but may require software framework support to achieve them automatically.

**Implications for Architecture Search**    Model specialization closely relates to model architecture search, where neural network architectures are automatically explored (e.g., via reinforcement learning (Zoph & Le (2017)). While architecture search has been typically employed to obtain more accurate networks for generic tasks (e.g., ImageNet and CIFAR), model specialization can be viewed as a form of architecture search for i) inference-efficient models for ii) a *specific* task. We believe it is promising to use inference speed as a means of guiding architecture search by searching for increasingly computationally-intensive architectures (e.g., stacking "cells" (Zoph et al., 2017)). Given that the Pareto frontier between accuracy and speed is often task-specific, incorporating task awareness in search is likely to further improve search speed and model efficiency.

