# OpenReview forum: "Model Specialization for Inference Via End-to-End Distillation, Pruning, and Cascades"
_ICLR.cc/2018/Conference — Reject_

### Official Review · AnonReviewer2 · 2017-11-23
**Good paper. Experiments can be improved.**

**Rating:** 6
**Confidence:** 3

**Review:**

The paper presents an approach to do task aware distillation, task-specific pruning and specialized cascades. The main result is that such methods can yield smaller, efficient and sometimes more accurate models.

The proposed approach is simple and easy to understand. The task aware distillation relies on the availability of data that is target specific. In practice, I believe this is not an unreasonable requirement.

The speedups and accuracy gains of this paper are impressive. The fact that the proposed technique is simple yet yields such speedups is encouraging. However, evaluating on simple datasets like Kaggle cat/dog and Oxford Flowers diminishes the value of the paper. I would strongly encourage the authors to try harder datasets such as COCO, VOC etc. This will make the paper more valuable to the community.

Missing citations
Do Deep Nets Really Need to be Deep? - Ba & Caruana 2014

---

### Official Review · AnonReviewer3 · 2017-11-27
**Weak Contribution and Confusing Experiments**

**Rating:** 4
**Confidence:** 4

**Review:**

This paper presents three different techniques for model specialization, i.e. adapting a pretrained network to a more specific task and reduce its computational cost while maintaining the performance. The three techniques are distillation, weight pruning and cascades. Evaluation compares how effective each technique is and how they interact with each other. In certain settings the obtained speed-up reaches 5x without loss of accuracy.

Pros:
- The idea of reducing the computational cost of specialized models makes sense.
- In some setting the speed-up can reach more than 5x, which is quite relevant.

Cons:
- The fact that the models are specialized to simpler tasks is not explicitly used in the approach. The authors should test what would happen when using their cascade for classification on all classes of ImageNet for instance. Would it be the gain in speed much lower?
- It is not clear if the distillation on smaller networks is really improving the models accuracy. The authors compared the distilled models with models trained from scratch. There should be an additional experiment with the small models trained on Imagenet first and then fine-tuned to the task. If in that case there is non gain, then, what is the advantage of distilling in these settings? ImageNet annotations need to be used anyway to train the teacher network.
- In section 3.2 it seems that the filters of a CNN are globally ranked based on their average activation values. Those with the lowest average activation will be removed. However, in my understanding, the ranking can work better if performed layer specific and not globally.
- In section 3.4, the title says "end-to-end specialization pipeline", but actually, the specialization is done in 3 steps, therefore in my understanding it is not end-to-end.
- There are some spelling errors, for instance in the beginning of section 4.1
- Pruning does not seem to produce much speed-up.
- The experimental part is difficult to read. In particular Fig. 4 should be better explained. There are some symbols in the legend that do not appear in the graph, and others (baselines only) that appear multiple times, but it is not clear what they represent. Also, at the end of the explanation of Fig. 4 the authors mention a gain of 8%, which in my understanding is not really relevant compared with the total speed-up, which can be in the order of 500%

Overall, the idea of model specialization seem interesting. However, in my understanding the main source of speed-up is a cascade approach with a reduced model, in which is not clear how much speed-up is actually due to the specialized task.

---

### Official Review · AnonReviewer1 · 2017-11-27
**Useful practical idea but flawed execution provides little value**

**Rating:** 3
**Confidence:** 4

**Review:**

The authors review and evaluate several empirical methods to create faster versions of big neural nets for vision without sacrificing accuracy. They show using the ResNet architecture that combining distillation, pruning, and cascades are complementary and can yield pretty nice speedups.

This is a great idea and could be a strong paper, but it's really hard to glean useful recommendations from this for several reasons:

- The writing of the paper makes it hard to understand exactly what's being compared and evaluated. For a paper like this it's really crucial to be precise. When the authors say "specialization" or "specialized model", they sometimes mean distillation, sometimes filter pruning, and sometimes cascades. The distinction of "task-aware" also seems arbitrary to me and obfuscates the contribution of the paper as well. As far as I can tell, the technique is exactly the same, all that's changing is a slight modification. It's not like any of the intuitions or objectives are changing, so adding this new terminology just complicates things. For example, just say "We distill a parent model to a child model with a subset of the labels."

- In terms of substance, the experiments don't really add much value in terms of general lessons. For example, the Cat/Dog from ImageNet distillation only works if the target labels are exactly a subset of the original. Obviously if the parent model was overcomplete before, it is certainly overcomplete now. The proposed cascade method is also fairly trivial -- a cheap distilled model backs off to the reference model. Why not train the whole cascade end-to-end? What about multiple levels of cascades? The only useful conclusion I can draw from the experiments is that (1) distillation still works (2) cascades also still work (3) pruning doesn't seem that useful in comparison. Training a cascade also involves a bunch of non-trivial design choices which are largely ignored -- how to set pass through, how to train the model, etc. etc.

- Nit: where are the blue squares in Figure 4? (Distill only) shouldn't those be the fastest methods (aside from pruning)?

An ideal story would for a paper like this would be: here are some complementary ideas that we can combine in non-obvious ways for superlinear benefits, e.g. it turns out that by distilling into a cascade in some end-to-end fashion, you can get much better accuracy vs. speed trade-offs. Instead this paper is a grab-back of tricks. Such a paper can also provide value, but to do that right, the tricks need to be obvious *in retrospect only* and/or the experiments need to show a lot of precise practical lessons. All in all this paper reads like a tech report but not a conference publication.

---

### Decision · Program_Chairs · 2018-01-29
**ICLR 2018 Conference Acceptance Decision**

**Decision:**

Reject

**Comment:**

This paper does not meet the bar for ICLR - neither in terms of the quality of the write-up, nor in experimental design. The two confident reviewers agree to reject the paper, the weak accept comes from a less confident reviewer who did not write a good review at all. The rebuttal does not change this assessment.